# Identification of novel antiviral host factors by functional gene expression analysis using *in vitro* HBV infection assay systems

Takuto Nosaka[1], Tatsushi Naito[1], Yu Akazawa[1], Kazuto Takahashi[1], Hidetaka Matsuda[1], Masahiro Ohtani[1], Tsutomu Nishizawa[2], Hiroaki Okamoto[2], Yasunari Nakamoto[1]*

1 Second Department of Internal Medicine, Faculty of Medical Sciences, University of Fukui, Fukui, Japan,
2 Division of Virology, Department of Infection and Immunity, Jichi Medical University School of Medicine, Tochigi, Japan

* nakamoto-med2@med.u-fukui.ac.jp

## Abstract

To cure hepatitis B virus (HBV) infection, it is essential to elucidate the function of hepatocyte host factors in regulating the viral life cycle. Signaling and transcription activator of transcription (STAT)1 play important roles in immune responses, but STAT1-independent pathways have also been shown to have important biological reactivity. Using an *in vitro* HBV infection assay system, the current study aimed to investigate the STAT1-independent host factors that contribute to the control of viral infection by comprehensive functional screening. The *in vitro* HBV infection system was established using primary human hepatocytes (PXB cells) infected with HBV derived from a plasmid containing the 1.3-mer HBV genome. Comprehensive functional studies were performed using small interfering RNA (siRNA) and vector transfection and analyzed using microarrays. Knockdown of STAT1 increased viral products in HBV-transfected HepG2 cells, but decreased in HBV-infected PXB cells. RNA microarray was performed using HBV-infected PXB cells with STAT1 knockdown. Fumarylacetoacetate hydrolase (FAH) was extracted by siRNA of genes in PXB cells altered by STAT1 knockdown. Transfection of FAH inhibited HBV replication. Dimethyl fumarate (DMF), the methyl ester of FAH metabolite, showed antiviral effects by inducing autophagy and anti-HBV-related genes. Independently of STAT1, FAH was identified as a host factor that contributes to the control of viral infection, and its metabolite, DMF, exhibited antiviral activity. These results suggest that the novel host factor FAH and its metabolites may be an innovative therapeutic strategy to control the HBV life cycle.

## Introduction

Chronic hepatitis B (CHB), caused by the hepatitis B virus (HBV), is a leading cause of liver fibrosis, cirrhosis, and hepatocellular carcinoma (HCC) worldwide [1,2]. HCC is currently the second leading cause of death from cancer, and more than 50% of HCC cases are linked to HBV infection in the most affected areas [3]. Current treatments for CHB include nucleos(t)ide analogues (NUCs) and pegylated interferon-γ (PEG-IFN-γ) [4]. Although NUCs can

**Data availability statement:** The raw microarray data were deposited in the NCBI Gene Expression Omnibus (GEO) under the accession number GSE253496.

**Funding:** This research was partially supported by Japan Agency for Medical Research and Development (AMED) under Grant Numbers JP23fk0210104 and JP23fk0210113, and Japan Society for the Promotion of Science (JSPS) KAKENHI Grant-in-Aid for Scientific Research Number 22K15992. The funders had no role in study design, data collection and analysis, decision to publish, or preparation of the manuscript.

**Competing interests:** The authors have declared that no competing interests exist.

**Abbreviations:** APOBEC, apolipoprotein B mRNA editing enzyme catalytic subunit; ATG, autophagy-related gene; cccDNA, covalently closed circular DNA; CHB, chronic hepatitis B; DMF, dimethyl fumarate; FAH, fumarylacetoacetate hydrolase; HBsAg, hepatitis B surface antigen; HBV, hepatitis B virus; HCC, hepatocellular carcinoma; IFN, interferon; IRF, interferon regulatory factor; JAK, Janus kinase; MOI, multiplicity of infection; MTT, 3-(4,5-di-methylthiazol-2-yl)-2,5-diphenyltetrazolium bromide; NES, normalized enrichment score; NNMT, nicotinamide N-methyltransferase; NRF, nuclear erythroid-related factor; NUC, nucleotide/nucleoside analog; PEG-IFN-γ, pegylated interferon-γ; PXB, primary human hepatocyte; qRT-PCR, quantitative real-time polymerase chain reaction; SCID, severe combined immunodeficiency; siRNA, small interfering RNA; SQSTM1, sequestosome 1; STAT; signal transducer and activator of transcription; uPA, urokinase-type plasminogen activator.

significantly suppress HBV DNA, they do not act directly on covalently closed circular DNA (cccDNA), the intranuclear template for HBV replication, and long-term treatment is usually required to maintain HBV suppression. PEG-IFN-γ-based therapies have the potential to cure infections, but suffer from low response rates and severe side effects.

Targeting the hepatocyte host factors involved in the viral life cycle may be a promising therapeutic approach to overcome cccDNA persistence. Among the host factors reported to be associated with the HBV life cycle, the signal transducer and activator of transcription (STAT) 1 protein plays a key role in the immune response by transducing signals from type I–III interferons (IFNs) [5–7]. It has been established that the major biological responses to IFN-γ are gene products regulated by the Janus kinase (JAK)-STAT pathway [8]. However, comprehensive screening, including microarray analysis using STAT1-deficient cells, has revealed that STAT-independent pathways also play an important role in the biological response to IFN-γ [9]. Elucidating the function of host factors associated with the HBV life cycle provides not only a basic understanding of HBV infection, but also has the potential to identify new antiviral targets and facilitate the development of new therapeutic strategies [10].

Using an *in vitro* HBV infection assay system [11, 12], this study was designed to investigate the STAT1-independent anti-HBV mechanisms in the HBV life cycle. Although the antiviral activity of STAT1 was seen in HepG2 cells transfected with the HBV genome, the viral products were decreased by small interfering RNA (siRNA) knockdown experiments of the STAT1 molecule in the infection system. Furthermore, comprehensive functional screening identified fumarylacetoacetate hydrolase (FAH) as a previously unidentified candidate that contributes to the control of viral infection independently of STAT1. Moreover, fumarate, a tyrosine metabolite produced by FAH, was shown to exhibit anti-HBV effects by inducing autophagy and anti-HBV-related genes in hepatocytes.

## Materials and methods

### Cell line

The human liver cancer cell line HepG2 was obtained from the American Type Culture Collection (Manassas, VA, USA) and cultured at 37°C with 5% $CO_2$ and RPMI-1640 (Sigma-Aldrich, St. Louis, MO, USA) containing 10% fetal bovine serum (FBS), 2 mmol/L L-glutamine, 1 μmol/L sodium pyruvate, 0.1 mmol/L nonessential amino acids, and 100 U/mL penicillin/50 μg/mL streptomycin (Gibco, Grand Island, NY, USA) [12].

### Primary human hepatocytes

Primary human hepatocytes (PXB cells) derived from chimeric urokinase-type plasminogen activator/severe combined immunodeficiency (uPA/SCID) mice with humanized livers were purchased from PhoenixBio Co., Ltd. (Hiroshima, Japan) and cultured in BioCoat Collagen I white plates (Corning Life Science, Tewksbury, MA, USA) using maintenance medium, as described previously [11,12].

### HBV plasmid transfection into HepG2 cells and treatment

The 1.3-mer HBV genome [genotype C2, both basal core promoter (BCP) A1762T/G1764A mutation and precore G1896A mutation, accession number AB819615] was inserted into the KpnI and NotI restriction sites of pBluescript II SK (Agilent Technologies, Santa Clara, USA). Subsequently, the recombinant HBV plasmid was introduced into HepG2 cells through transfection using Lipofectamine LTX Reagent (Thermo Fisher Scientific,

Waltham, MA, USA) [12]. HepG2 cells transfected with HBV, exhibiting stable production of hepatitis B surface antigen (HBsAg) in the supernatant, were identified as HepG2. D11 clone. Dimethyl fumarate (DMF, Sigma-Aldrich, 242926) was added to the medium at different concentrations.

### In vitro HBV infection to primary human hepatocyte

HBV was collected from the supernatant of HepG2 cells transfected with HBV plasmids and digested with recombinant DNase I (Takara Bio, Shiga, Japan). HBV DNA was extracted using SMI TEST EX-R&D (Medical & Biological Laboratories Co., Ltd., Nagano, Japan), and the amount was determined using real-time quantitative polymerase chain reaction (qPCR). PXB cells were infected with HBV at a multiplicity of infection (MOI) of 500 for 24 h in the presence of 4% PEG 8000 (Sigma-Aldrich) [12].

### Interferon treatment

IFN-γ (PeproTech, Rocky Hill, NJ, USA) was diluted in culture medium and added to PXB cells and HepG2.D11 cells at the concentrations of $5 \times 10^3$ ng/mL as described previously [11].

### Quantification of HBsAg

The HBsAg levels in the supernatant were determined using Lumipulse HBsAg-HQ immuno-assay (Fujirebio Inc., Tokyo, Japan) [12].

### RNA and DNA extraction and cDNA synthesis

Total RNA was extracted using TRIzol reagent (Thermo Fisher Scientific) and cDNA synthesis was performed using a High-Capacity RNA-to-cDNA Kit (Applied Biosystems, Foster City, CA, USA). DNA was extracted from the supernatant and HepG2.D11 and PXB cells using SMITEST EX-R&D (MBL) [12].

### Quantification of HBV DNA and HBV RNA

Two pairs of primers, corresponding to the DNA regions of the HBV genome, were used for the assay (Table 1). Nested PCR was performed for HBV DNA, and the quantitative gene expression levels of HBV DNA and RNA were determined by real-time PCR using the StepOne Plus system (Applied Biosystems) [12]. Primers and probes were obtained from Takara Bio and Applied Biosystems (Table 1). Glyceraldehyde-3-phosphate dehydrogenase (GAPDH; Applied Biosystems) and transferrin receptor (TFRC; Sigma-Aldrich) were used as endogenous controls [12]. The conversion of HBV DNA was performed by the following equation: 1 pg/ml = $2.83 \times 10^5$ copies/ml = 5.45 log10 copies/ml [13,14].

**Table 1. Nucleotide sequences of primers and a probe used for real-time quantitative polymerase chain reaction.**

| Primer name | | Nucleotide sequence of primers and a probe (5'-3') | Nucleotide position in HBV |
|---|---|---|---|
| HBV DNA, RNA-1816F | Forward | GCAACTTTTTCACCTCTGCCTA | 1816-1837 |
| HBV DNA, RNA-1974R | Reverse | GGAAAGAAGTCAGAAGGCAA | 1974-1955 |
| HBV DNA, RNA-1826F | Forward | CACCTCTGCCTAATCATC | 1826-1843 |
| HBV DNA, RNA-1947R | Reverse | AGTAACTCCACAGTAGCTCCAAATT | 1947-1923 |
| HBV DNA, RNA-Probe | Probe | (FAM)-TTCAAGCCTCCAAGCTGTGCCTTG-(TAMRA) | 1863-1886 |

Nucleotide sequence of the primers and a probe

| Gene name | | Nucleotide sequence of primers and a probe (5'-3') |
|---|---|---|
| Transferrin receptor | Forward | GGACACCTATAAGGAACTGATTGAGA |
| | Reverse | AGTCCAGGTTCAATTCAACATCATG |
| | Probe | (FAM)-AATCACGAACTGACCAGCGACCTCTGC-(TAMRA) |

## Quantitative gene expression analysis

Gene expression was analyzed with custom TaqMan Array plate and TaqMan Array 96-well plate, fast (Thermo Fisher Scientific) and the selected target genes were determined using the StepOne Plus real-time PCR system (Applied Biosystems) (Table 2). Expression levels of the target genes were analyzed using the ΔΔCt comparative threshold method. The GAPDH gene was used as an internal control.

**Table 2. Primers used in this study. Primers used in custom TaqMan® Array Fast plate.**

| | Gene Symbol | Species | Dye | Assay ID | Company |
|---|---|---|---|---|---|
| 1 | 18s rRNA | Human | FAM | Hs99999901_s1 | Applied Biosystems |
| 2 | ABCC1 | Human | FAM | Hs01561483_m1 | Applied Biosystems |
| 3 | ABCC2 | Human | FAM | Hs00960489_m1 | Applied Biosystems |
| 4 | ABCG2 | Human | FAM | Hs01053790_m1 | Applied Biosystems |
| 5 | ACOX1 | Human | FAM | Hs01074241_m1 | Applied Biosystems |
| 6 | ACTB | Human | FAM | Hs99999903_m1 | Applied Biosystems |
| 7 | ADSL | Human | FAM | Hs01075807_m1 | Applied Biosystems |
| 8 | AHR | Human | FAM | Hs00169233_m1 | Applied Biosystems |
| 9 | AKR1C1 | Human | FAM | Hs04230636_sH | Applied Biosystems |
| 10 | APOBEC3A | Human | FAM | Hs02572821_s1 | Applied Biosystems |
| 11 | APOBEC3B | Human | FAM | Hs00358981_m1 | Applied Biosystems |
| 12 | APOBEC3G | Human | FAM | Hs00222415_m1 | Applied Biosystems |
| 13 | ASL | Human | FAM | Hs00902699_m1 | Applied Biosystems |
| 14 | ATG5 | Human | FAM | Hs00355494_m1 | Applied Biosystems |
| 15 | ATG7 | Human | FAM | Hs00893766_m1 | Applied Biosystems |
| 16 | BACH1 | Human | FAM | Hs00230917_m1 | Applied Biosystems |
| 17 | BCL2 | Human | FAM | Hs04986394_s1 | Applied Biosystems |
| 18 | BCL2L1 | Human | FAM | Hs00236329_m1 | Applied Biosystems |
| 19 | BDH1 | Human | FAM | Hs00366297_m1 | Applied Biosystems |
| 20 | BLVRA | Human | FAM | Hs00167599_m1 | Applied Biosystems |
| 21 | BTRC | Human | FAM | Hs00182707_m1 | Applied Biosystems |
| 22 | CASP1 | Human | FAM | Hs00354836_m1 | Applied Biosystems |
| 23 | CAT | Human | FAM | Hs00156308_m1 | Applied Biosystems |
| 24 | CCL5 | Human | FAM | Hs00982282_m1 | Applied Biosystems |
| 25 | CHUK | Human | FAM | Hs00989502_m1 | Applied Biosystems |
| 26 | CXCL9 | Human | FAM | Hs00171065_m1 | Applied Biosystems |
| 27 | FAH | Human | FAM | Hs00164611_m1 | Applied Biosystems |
| 28 | FH | Human | FAM | Hs00264683_m1 | Applied Biosystems |
| 29 | FTH1 | Human | FAM | Hs01694011_s1 | Applied Biosystems |
| 30 | G6PD | Human | FAM | Hs00166169_m1 | Applied Biosystems |
| 31 | GAPDH | Human | FAM | Hs99999905_m1 | Applied Biosystems |
| 32 | GCLC | Human | FAM | Hs00155249_m1 | Applied Biosystems |

*(Continued)*

Table 2. (Continued)

| | Gene Symbol | Species | Dye | Assay ID | Company |
|---|---|---|---|---|---|
| 33 | GCLM | Human | FAM | Hs00978072_m1 | Applied Biosystems |
| 34 | GPX1 | Human | FAM | Hs00829989_gH | Applied Biosystems |
| 35 | GPX2 | Human | FAM | Hs01591589_m1 | Applied Biosystems |
| 36 | GSR | Human | FAM | Hs00167317_m1 | Applied Biosystems |
| 37 | GSTA1 | Human | FAM | Hs00275575_m1 | Applied Biosystems |
| 38 | GSTM1 | Human | FAM | Hs01683722_gH | Applied Biosystems |
| 39 | GSTP1 | Human | FAM | Hs00943350_g1 | Applied Biosystems |
| 40 | GSTZ1 | Human | FAM | Hs01041668_m1 | Applied Biosystems |
| 41 | HGD | Human | FAM | Hs01056732_m1 | Applied Biosystems |
| 42 | HIF1A | Human | FAM | Hs00153153_m1 | Applied Biosystems |
| 43 | HMOX1 | Human | FAM | Hs01110250_m1 | Applied Biosystems |
| 44 | HPD | Human | FAM | Hs00157976_m1 | Applied Biosystems |
| 45 | IDH1 | Human | FAM | Hs04966975_g1 | Applied Biosystems |
| 46 | IFI16 | Human | FAM | Hs00194261_m1 | Applied Biosystems |
| 47 | IFI44L | Human | FAM | Hs00199115_m1 | Applied Biosystems |
| 48 | IFNA1 | Human | FAM | Hs00855471_g1 | Applied Biosystems |
| 49 | IFNA2 | Human | FAM | Hs00265051_s1 | Applied Biosystems |
| 50 | IFNAR1 | Human | FAM | Hs01066118_m1 | Applied Biosystems |
| 51 | IFNG | Human | FAM | Hs00174143_m1 | Applied Biosystems |
| 52 | IFNGR1 | Human | FAM | Hs00166223_m1 | Applied Biosystems |
| 53 | IKBKB | Human | FAM | Hs00233287_m1 | Applied Biosystems |
| 54 | IKBKE | Human | FAM | Hs01063858_m1 | Applied Biosystems |
| 55 | IL1B | Human | FAM | Hs01555410_m1 | Applied Biosystems |
| 56 | IRF3 | Human | FAM | Hs01547283_m1 | Applied Biosystems |
| 57 | IRF7 | Human | FAM | Hs01014809_g1 | Applied Biosystems |
| 58 | IRF9 | Human | FAM | Hs00196051_m1 | Applied Biosystems |
| 59 | JAK1 | Human | FAM | Hs01026983_m1 | Applied Biosystems |
| 60 | JAK2 | Human | FAM | Hs01078124_m1 | Applied Biosystems |
| 61 | KEAP1 | Human | FAM | Hs00202227_m1 | Applied Biosystems |
| 62 | LIPH | Human | FAM | Hs00975890_m1 | Applied Biosystems |
| 63 | MAF | Human | FAM | Hs04185012_s1 | Applied Biosystems |
| 64 | ME1 | Human | FAM | Hs00159110_m1 | Applied Biosystems |
| 65 | ME2 | Human | FAM | Hs00929809_g1 | Applied Biosystems |
| 66 | NFKB1 | Human | FAM | Hs00765730_m1 | Applied Biosystems |
| 67 | NLRP3 | Human | FAM | Hs00918082_m1 | Applied Biosystems |
| 68 | NOTCH1 | Human | FAM | Hs01062014_m1 | Applied Biosystems |
| 69 | NQO1 | Human | FAM | Hs01045993_g1 | Applied Biosystems |
| 70 | OAS2 | Human | FAM | Hs00942643_m1 | Applied Biosystems |
| 71 | OSGIN1 | Human | FAM | Hs00203539_m1 | Applied Biosystems |
| 72 | PGD | Human | FAM | Hs00427230_m1 | Applied Biosystems |
| 73 | PLA2G7 | Human | FAM | Hs00965837_m1 | Applied Biosystems |
| 74 | POMP | Human | FAM | Hs01106088_m1 | Applied Biosystems |
| 75 | PRDX1 | Human | FAM | Hs00602020_mH | Applied Biosystems |
| 76 | PSMA1 | Human | FAM | Hs01027360_g1 | Applied Biosystems |
| 77 | PSMB5 | Human | FAM | Hs00605652_m1 | Applied Biosystems |
| 78 | RAD51 | Human | FAM | Hs00947967_m1 | Applied Biosystems |
| 79 | RELA | Human | FAM | Hs00153294_m1 | Applied Biosystems |

*(Continued)*

**Table 2.** (Continued)

|  | Gene Symbol | Species | Dye | Assay ID | Company |
|---|---|---|---|---|---|
| 80 | RXRA | Human | FAM | Hs01067640_m1 | Applied Biosystems |
| 81 | SDHA | Human | FAM | Hs07291714_mH | Applied Biosystems |
| 82 | SETD2 | Human | FAM | Hs00383442_m1 | Applied Biosystems |
| 83 | SIRT1 | Human | FAM | Hs01009006_m1 | Applied Biosystems |
| 84 | SLC7A11 | Human | FAM | Hs00921938_m1 | Applied Biosystems |
| 85 | SOD1 | Human | FAM | Hs00533490_m1 | Applied Biosystems |
| 86 | SOD2 | Human | FAM | Hs00167309_m1 | Applied Biosystems |
| 87 | SQSTM1 | Human | FAM | Hs01061917_g1 | Applied Biosystems |
| 88 | SRXN1 | Human | FAM | Hs00607800_m1 | Applied Biosystems |
| 89 | STAT1 | Human | FAM | Hs01013996_m1 | Applied Biosystems |
| 90 | STAT2 | Human | FAM | Hs01013123_m1 | Applied Biosystems |
| 91 | SYVN1 | Human | FAM | Hs00381211_m1 | Applied Biosystems |
| 92 | TAT | Human | FAM | Hs00356930_m1 | Applied Biosystems |
| 93 | TNF | Human | FAM | Hs01113624_g1 | Applied Biosystems |
| 94 | TXN | Human | FAM | Hs01555214_g1 | Applied Biosystems |
| 95 | TXNRD1 | Human | FAM | Hs00917067_m1 | Applied Biosystems |
| 96 | ULK1 | Human | FAM | Hs00177504_m1 | Applied Biosystems |

Primers used in this study

| | Gene Symbol | Species | Dye | Assay ID | Company |
|---|---|---|---|---|---|
| | STAT1 | Human | FAM | Hs01013996_m1 | Applied Biosystems |
| | IRF2 | Human | FAM | Hs01082884_m1 | Applied Biosystems |
| | IFI44L | Human | FAM | Hs00915292_m1 | Applied Biosystems |
| | FAH | Human | FAM | Hs00164611_m1 | Applied Biosystems |
| | NNMT | Human | FAM | Hs00196287_m1 | Applied Biosystems |
| | NFE2L2 | Human | FAM | Hs00975961_g1 | Applied Biosystems |

## Microarray analysis

Microarray analyses were performed by Takara Bio using the SurePrint G3 Human GE 8 × 60 K v3 Microarray (Agilent Technologies), as described previously [11, 12]. The microarray data were validated and normalized using GeneSpring ver. 14.9.1 software (Agilent Technologies) by Hokkaido System Science Co., Ltd. (Hokkaido, Japan). The median shift normalization to 75 percentile and baseline transformation using the median of the control samples were applied. Probes with 100.0% of samples in any one of two conditions were flagged as [Detected] and gene expression values less than the cut-off value (10.0) were excluded to remove spots with low signal values and low reliability. The raw microarray data were deposited in the NCBI Gene Expression Omnibus (GEO) under the accession number GSE253496 (https://www.ncbi.nlm.nih.gov/geo/query/acc.cgi?acc=GSE253496). Functional annotation was performed on the samples using the gene set enrichment analysis (GSEA) method [15], supported by the Broad Institute website (https://www.gsea-msigdb.org/gsea/index.jsp) and performed using GSEA software (version 4.3.2; Broad Institute, Inc., Massachusetts Institute of Technology, Boston, MA, USA, and Regents of the University of California, CA, USA). False-discovery rate (FDR) q-values < 0.25 and adjusted p values < 0.05 were considered significant enrichment.

## Small interfering RNA

siRNA constructs were obtained using siGENOME SMARTpool reagents (Horizon Discovery, Lafayette, CO, USA) that targeted siGENOME SMARTpool and siGENOME Non-Targeting Control siRNA (Table 3) [12]. We used an siGENOME SMARTpool Cherry-pick library (Horizon Discovery) of 43 candidate genes to screen for RNA interference (Table 3). PXB cells and HepG2. D11 cells were transfected with 25 nM siRNA using DharmaFECT 4 transfection reagent (Horizon Discovery).

**Table 3. Catalog number and descriptions lists of siRNA.**

| Pool Catalog Number | Gene Symbol | Gene Accession | Description |
|---|---|---|---|
| M-003543-01 | STAT1 | 6772 | STAT1 siGENOME SMART - Human |
| M-011705-01 | IRF2 | 3660 | IRF2 siGENOME SMART - Human |
| M-004599-00 | IFI44L | 10964 | IFI44L siGENOME SMART - Human |
| D-001206-13 | | | siGENOME Non-Targeting Control siRNA Pool #1 |

Cherry-Pick Custom Library (siRNA)

| | Pool Catalog Number | Gene Symbol | Gene Accession | Description |
|---|---|---|---|---|
| 1 | M-009821-00 | DPYSL3 | 1809 | DPYSL3 siGENOME SMARTpool - Human |
| 2 | M-012559-02 | IGFBP1 | 3484 | IGFBP1 siGENOME SMARTpool - Human |
| 3 | M-009635-00 | FAH | 2184 | FAH siGENOME SMARTpool - Human |
| 4 | M-008336-00 | TAT | 6898 | TAT siGENOME SMARTpool - Human |
| 5 | M-010855-01 | DDIT4 | 54541 | DDIT4 siGENOME SMARTpool - Human |
| 6 | M-014727-00 | RTP3 | 83597 | RTP3 siGENOME SMARTpool - Human |
| 7 | M-008506-00 | TDO2 | 6999 | TDO2 siGENOME SMARTpool - Human |
| 8 | M-009910-01 | HSD11B1 | 3290 | HSD11B1 siGENOME SMARTpool - Human |
| 9 | M-009946-01 | GPAM | 57678 | GPAM siGENOME SMARTpool - Human |
| 10 | M-003410-01 | NR0B2 | 8431 | NR0B2 siGENOME SMARTpool - Human |
| 11 | M-008286-01 | CYP2C8 | 1558 | CYP2C8 siGENOME SMARTpool - Human |
| 12 | M-008822-01 | ATF5 | 22809 | ATF5 siGENOME SMARTpool - Human |
| 13 | M-003746-02 | MAF | 4094 | MAF siGENOME SMARTpool - Human |
| 14 | M-004819-03 | DDIT3 | 1649 | DDIT3 siGENOME SMARTpool - Human |
| 15 | M-003754-01 | TRIB3 | 57761 | TRIB3 siGENOME SMARTpool - Human |
| 16 | M-008169-00 | CYP3A4 | 1576 | CYP3A4 siGENOME SMARTpool - Human |
| 17 | M-008726-00 | AQP7 | 364 | AQP7 siGENOME SMARTpool - Human |
| 18 | M-004297-01 | SLC7A2 | 6542 | SLC7A2 siGENOME SMARTpool - Human |
| 19 | M-010351-01 | NNMT | 4837 | NNMT siGENOME SMARTpool - Human |
| 20 | M-007966-02 | IL1RN | 3557 | IL1RN siGENOME SMARTpool - Human |
| 21 | M-007443-01 | SLC22A1 | 6580 | SLC22A1 siGENOME SMARTpool - Human |
| 22 | M-003964-01 | DUSP6 | 1848 | DUSP6 siGENOME SMARTpool - Human |
| 23 | M-004777-01 | IGFBP3 | 3486 | IGFBP3 siGENOME SMARTpool - Human |
| 24 | M-009703-00 | OGDHL | 55753 | OGDHL siGENOME SMARTpool - Human |
| 25 | M-022006-01 | EXOC3L4 | 91828 | EXOC3L4 siGENOME SMARTpool - Human |
| 26 | M-017077-00 | PPP1R3C | 5507 | PPP1R3C siGENOME SMARTpool - Human |
| 27 | M-021435-00 | INHBE | 83729 | INHBE siGENOME SMARTpool - Human |
| 28 | M-015771-00 | G0S2 | 50486 | G0S2 siGENOME SMARTpool - Human |
| 29 | M-018337-01 | MT1X | 4501 | MT1X siGENOME SMARTpool - Human |

| | Pool Catalog Number | Gene Symbol | Gene Accession | Description |
|---|---|---|---|---|
| 30 | M-003265-01 | FOS | 2353 | FOS siGENOME SMARTpool - Human |
| 31 | M-003723-02 | IKBKE | 9641 | IKBKE siGENOME SMARTpool - Human |
| 32 | M-011179-00 | GSTP1 | 2950 | GSTP1 siGENOME SMARTpool - Human |
| 33 | M-010723-02 | S100A14 | 57402 | S100A14 siGENOME SMARTpool - Human |
| 34 | M-027199-01 | SALL2 | 6297 | SALL2 siGENOME SMARTpool - Human |
| 35 | M-020868-01 | DCDC2 | 51473 | DCDC2 siGENOME SMARTpool - Human |
| 36 | M-013833-03 | ANGPTL8 | 55908 | ANGPTL8 siGENOME SMARTpool - Human |
| 37 | M-009324-01 | TBXAS1 | 6916 | TBXAS1 siGENOME SMARTpool - Human |
| 38 | M-004529-02 | NCF2 | 4688 | NCF2 siGENOME SMARTpool - Human |
| 39 | M-005472-02 | CXCR3 | 2833 | CXCR3 siGENOME SMARTpool - Human |
| 40 | M-003543-01 | STAT1 | 6772 | STAT1 siGENOME SMARTpool - Human |
| 41 | M-010995-01 | APOA4 | 337 | APOA4 siGENOME SMARTpool - Human |
| 42 | M-015096-01 | BEX1 | 55859 | BEX1 siGENOME SMARTpool - Human |
| 43 | M-025114-01 | HMGCLL1 | 54511 | HMGCLL1 siGENOME SMARTpool - Human |
| 44 | D-001206-13 | | | siGENOME Non-Targeting Control siRNA Pool #1 |

## Vector transfection

The cytomegalovirus vector used to overexpress STAT1 [pRP[Exp]-CMV > hSTAT1 (NM_001384891.1)] and FAH [pRP[Exp]-CMV > hFAH (NM_001374380.1)] was constructed and packaged using VectorBuilder. Further information is available on the VectorBuilder website (https://en.vectorbuilder.com/) under the VectorBuilder ID, STAT1: VB230117-1096kjt, FAH: VB230117-1091yve. The plasmid vector was transfected into HepG2.D11 cells using Lipofectamine LTX Reagent (Thermo Fisher Scientific).

## Fumarate assay

The abundance of intracellular fumarate was assessed using a fumarate colorimetric assay kit (Sigma-Aldrich, MAK060) that uses an enzyme assay. Briefly, cells ($1 \times 10^6$) were collected and homogenized in 100 µL of Fumarate Assay Buffer and centrifuged the sample at 13,000 ×g for 10 min to remove insoluble material. Absorbance was measured at 450 nm and compared to standard curves, according to the manufacturer's instructions.

## MTT assay

Cell viability was determined using the Cell Counting kit-8 (Dojindo, Kumamoto, Japan), a modified 3-(4,5-di-methylthiazol-2-yl)-2,5-diphenyltetrazolium bromide (MTT) assay. The optical density values of untreated and vehicle-treated cells were compared to obtain the ratios of cell numbers.

## Western blotting

HepG2.D11 cells were prepared using radioimmunoprecipitation assay (RIPA) lysis buffer (50 mmol/L Tris-HCl buffer [pH7.6], 150 mmol/L NaCl, 1% Nonidet® P40 Substitute, 0.5% Sodium Deoxycholate, 0.1% SDS) (Nakarai Tesuque, Kyoto, Japan). Anti-STAT1 antibody (Cell Signaling Technology, Danvers, MA, USA, #9175), anti-FAH (Sigma-Aldrich, HPA041370) antibody, and anti-β-actin (D6A8) monoclonal antibody (Cell Signaling Technology) were used for protein detection. Immune complexes were visualized using enhanced chemiluminescence detection reagents (Amersham Biosciences, Piscataway, NJ, USA) according to the manufacturer's protocol.

### Statistical analyses

Statistical analyses were performed using GraphPad Prism software (version 10; Graph-Pad Software Inc., San Diego, CA, USA). Statistical significance was determined using Mann-Whitney U tests or one-way analysis of variance followed by the Tukey–Kramer post-hoc test [12]. p values < 0.05 were considered statistically significant. Any comparisons not shown on graphs are non-significant.

## Results

### Knockdown of STAT1 increased viral products in HBV-transfected HepG2 cells, but decreased in the primary human hepatocyte infection system

siRNA transfection experiments were performed using an *in vitro* HBV infection assay system with PXB cells, together with the reported host immune-related genes, IRF2 [16] and IFI44L [12] (Fig 1A). IFN-γ significantly increased the expression of STAT1, IRF2, and IFI44L in PXB cells (Fig 1B). siRNA transfection decreased the expression of each target gene in the presence/absence of IFN-γ. Compared with IRF2 and IFI44L knockdown, STAT1 knockdown significantly reduced extracellular HBsAg and HBV DNA and intracellular HBV DNA and HBV RNA levels (Fig 1C). IFN-γ treatment significantly reduced HBV transfection levels, and knockdown of STAT1 further reduced HBV replication levels in the presence of IFN-γ. Next, we performed siRNA transfection experiments targeting the STAT1 gene using HepG2 cells transfected with HBV, exhibiting stable production of HBsAg and HBV in the supernatant (HepG2.D11 cells) (Fig 1D). Similar to PXB cells, IFN-γ significantly increased STAT1 gene expression in HepG2.D11 cells, and siRNA transfection decreased STAT1 expression (Fig 1E). In contrast to PXB cells, STAT1 knockdown in HepG2.D11 cells resulted in increased extracellular HBsAg and HBV DNA levels and intracellular of HBV DNA levels (Fig 1F). To overexpress STAT1 in HepG2.D11 cells, STAT1 vector was transfected (Fig 1G). Increased STAT1 expression was confirmed by quantitative real-time polymerase chain reaction (PCR) (Fig 1H) and western blot (Fig 1I). Overexpression of STAT1 decreased extracellular HBsAg and HBV DNA and intracellular HBV DNA and RNA levels (Fig 1J). These results indicate that the host gene STAT1 showed anti-HBV activity in conventional HBV-transfected HepG2 cells, whereas in the primary human hepatocyte HBV infection system, STAT1 knockdown conversely reduced HBV replication capacity.

### RNA microarray analysis in knockdown of STAT1 in primary human hepatocyte HBV infection system and HBV-transfected HepG2 cells

To investigate the mechanism by which STAT1 knockdown reduces HBV levels in primary human hepatocytes, RNA microarray analysis was performed to comprehensively examine the molecular changes in PXB cells. In HBV-infected PXB cells, STAT1 knockdown enriched 24 gene sets (p < 0.05, FDR q-value < 0.25) (Fig 2A). Of the 24 gene sets, 21 were associated with metabolism and protein synthesis. However, STAT1 knockdown in HepG2.D11 cells and IRF2 knockdown in PXB cells did not result in similar molecular changes (Fig 2B). From the 58,341 genes, 12,405 mRNAs were extracted after excluding genes with low signal values and low reliability [PXB cell si-Non-Target (Raw value) > 100] (Fig 2C). In HBV-infected PXB cells, STAT knockdown increased expression of 43 genes [si-STAT1/si-Non-Target (Log2 ratio) > 1] and decreased expression of 23 genes [si-STAT1/si-Non-Target (Log2 ratio) < -1] (Figs 2C–E). In HepG2.D11 cells, STAT knockdown increased expression of 340 genes [si-STAT1/si-Non-Target (Log2 ratio) > 1] and decreased expression of 266 genes [si-STAT1/si-Non-Target (Log2 ratio) < -1] (Fig 2D).

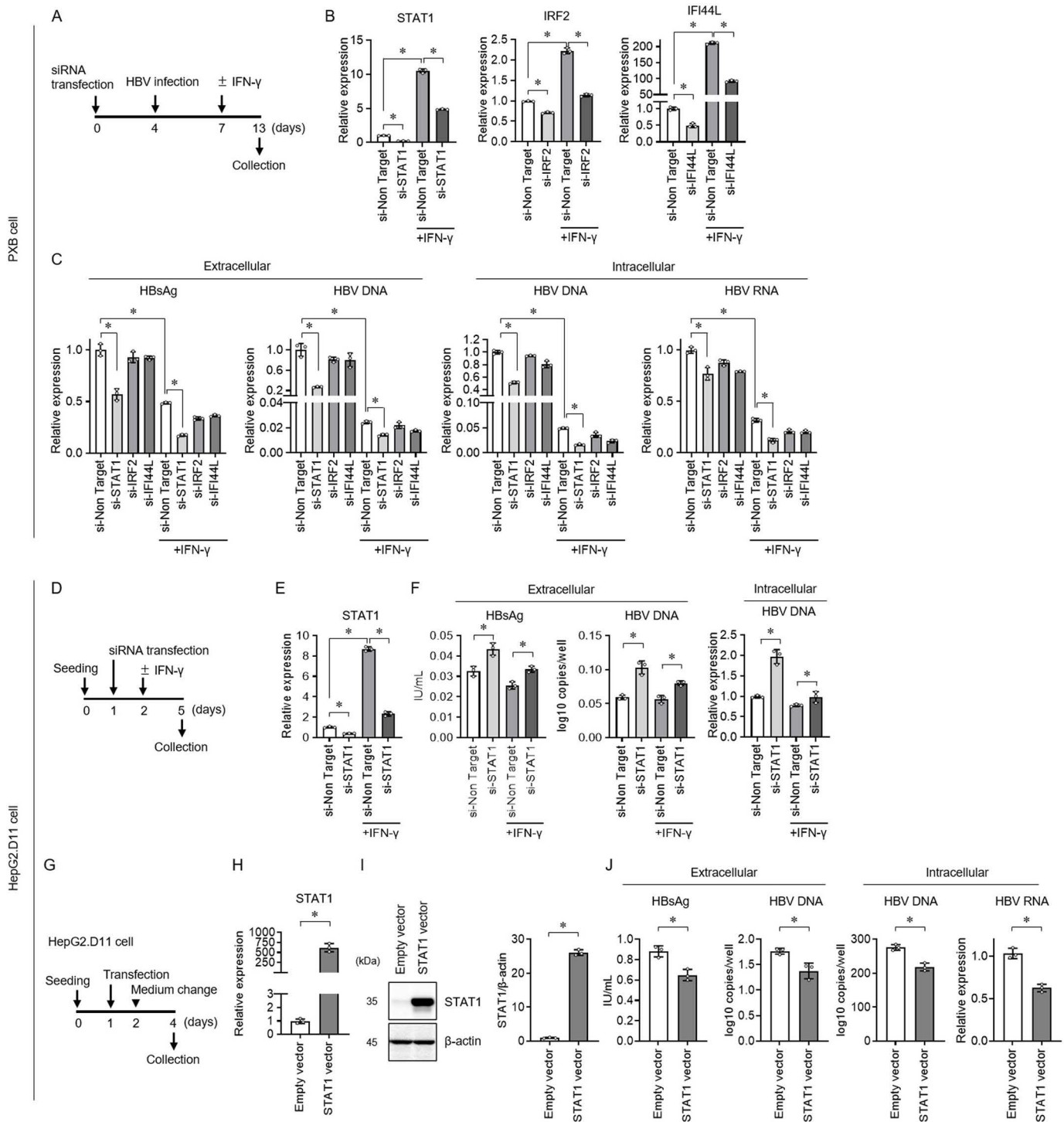

**Fig 1. Host factor gene knockdown analysis in primary human hepatocyte hepatitis B virus (HBV) infection system and HBV-transfected HepG2 cells.** (A) siRNA was transfected into the PXB cells on day 0 and HBV was added on day 4. IFN-γ was diluted and added to the medium on day 7 and the supernatant and PXB cells were collected on day 13. In the control group, only medium was added. The culture medium was exchanged on day 4, 6, and 7. (B) mRNA expression of STAT1, IRF2, and IFI44L, and (C) extracellular hepatitis B surface antigen (HBsAg) and HBV DNA and intracellular HBV DNA and HBV RNA were analyzed on day 13. (D) HepG2.D11 cells were seeded on day 0 and siRNA was transfected at medium change on day 1. IFN-γ was diluted and added to the medium on day 2 and the supernatant and HepG2.D11 cells were collected on day 5. In the control group, only medium was added. (E) mRNA expression of STAT1 and (F) extra-cellular HBsAg and HBV DNA and intracellular HBV DNA was analyzed on day 5. (G) HepG2.D11 cells were seeded on day 0 and CMV vector inserted cDNA STAT1 was transfected on day 1. The culture medium was exchanged on day 2 and the supernatant and HepG2.D11 cells were collected on day 4. (H) mRNA

expression of STAT1 and (I) western blot analysis of extracts from HepG2.D11 using STAT1 antibody and normalized to β-actin on day 4. (J) Extracellular HBsAg and HBV DNA and intracellular HBV DNA and HBV RNA were analyzed on day 4. Data are presented as the mean ± standard deviation (n = 3). (D-F) and (G-J) The experiments were replicated three times. *p < 0.05 using the Mann-Whitney U test. Abbreviations: HBsAg, hepatitis B surface antigen; HBV, hepatitis B virus; IFN, interferon; IRF, interferon regulatory factor; PXB, primary human hepatocyte; siRNA, small interfering RNA; STAT1, signal transducer and activator of transcription 1.

### A comprehensive functional screen identified FAH as a candidate host factor that regulates HBV infection

As STAT1 knockdown in HBV-infected PXB cells suppressed HBV replication levels, we investigated genes with anti-HBV activity among the candidate molecules. Forty-three genes that were highly expressed in PXB cells were identified using microarray analysis. siRNA transfection of the 43 genes was performed using HBV-infected PXB cells. siRNA transfection of 12 genes increased extracellular HBsAg levels by more than 1.2-fold (S1 Fig A). To identify candidate genes with a STAT1-independent anti-HBV effect, double siRNA transfection with STAT1 was performed on each of these 12 genes (S1 Fig B). With STAT1 knockdown, two genes, FAH and nicotinamide N-methyltransferase (NNMT), showed a more than 1.5-fold increase in HBsAg levels. In the STAT1 knockdown condition, knockdown of FAH significantly elevated intracellular HBV DNA. The knockdown efficiency was confirmed in two genes, FAH and NNMT (Fig 3A). Knockdown of FAH and NNMT significantly elevated extracellular HBsAg and intracellular HBV DNA levels (Fig 3B). FAH and NNMT siRNA transfection experiments were performed in HepG2.D11 cells (Figs 3C and D). Knockdown of FAH significantly elevated extracellular HBsAg, HBV DNA, intracellular HBV DNA, and HBV RNA levels and increased HBV replication levels compared to NNMT. These results suggest that FAH has anti-HBV effects, as determined by siRNA screening of genes altered by STAT1 knockdown in PXB cells. FAH knockdown resulted in increased HBV replication in the primary human hepatocyte infection system and conventional HBV-transfected HepG2 clone.

### Elevated FAH gene expression inhibits HBV replication

To overexpress FAH in HepG2.D11 cells, FAH cDNA was inserted into a plasmid vector and transfected (Fig 4A). Increased FAH expression was confirmed using qRT-pCR and western blot (Figs 4B and C). Overexpression of FAH resulted in decreased extracellular HBsAg and intracellular HBV DNA and RNA levels (Fig 4D). These results suggest that elevated FAH expression in hepatocytes inhibits HBV replication.

### Dimethyl fumarate exhibited antiviral effects by inducing autophagy and anti-HBV-related genes

FAH is the terminal step in the tyrosine catabolic pathway. The conversion of 4-fumarylacetoacetate to fumarate and acetoacetate is catalyzed by FAH [17] (Fig 5A). FAH knockdown in HepG2.D11 cells reduced intracellular fumarate levels (Fig 5B). Dimethyl fumarate (DMF), the methyl ester of fumaric acid, is a cell-permeable fumarate derivative [18] (Fig 5C). In HepG2.D11 cells, DMF decreased extracellular HBsAg and HBV DNA levels and intracellular HBV DNA and RNA levels (Fig 5D). No cytotoxicity was seen in DMF at concentrations of 0–30 μM (Fig 5E), and no difference in the morphology of cells was observed by microscopic examination (Fig 5F). To examine the association between DMF and the antiviral response, the expression of intracellular molecules was analyzed using qRT-PCR (Fig 5G and Table 2). Referring to papers related to DMF and antiviral functions, 96 molecules were

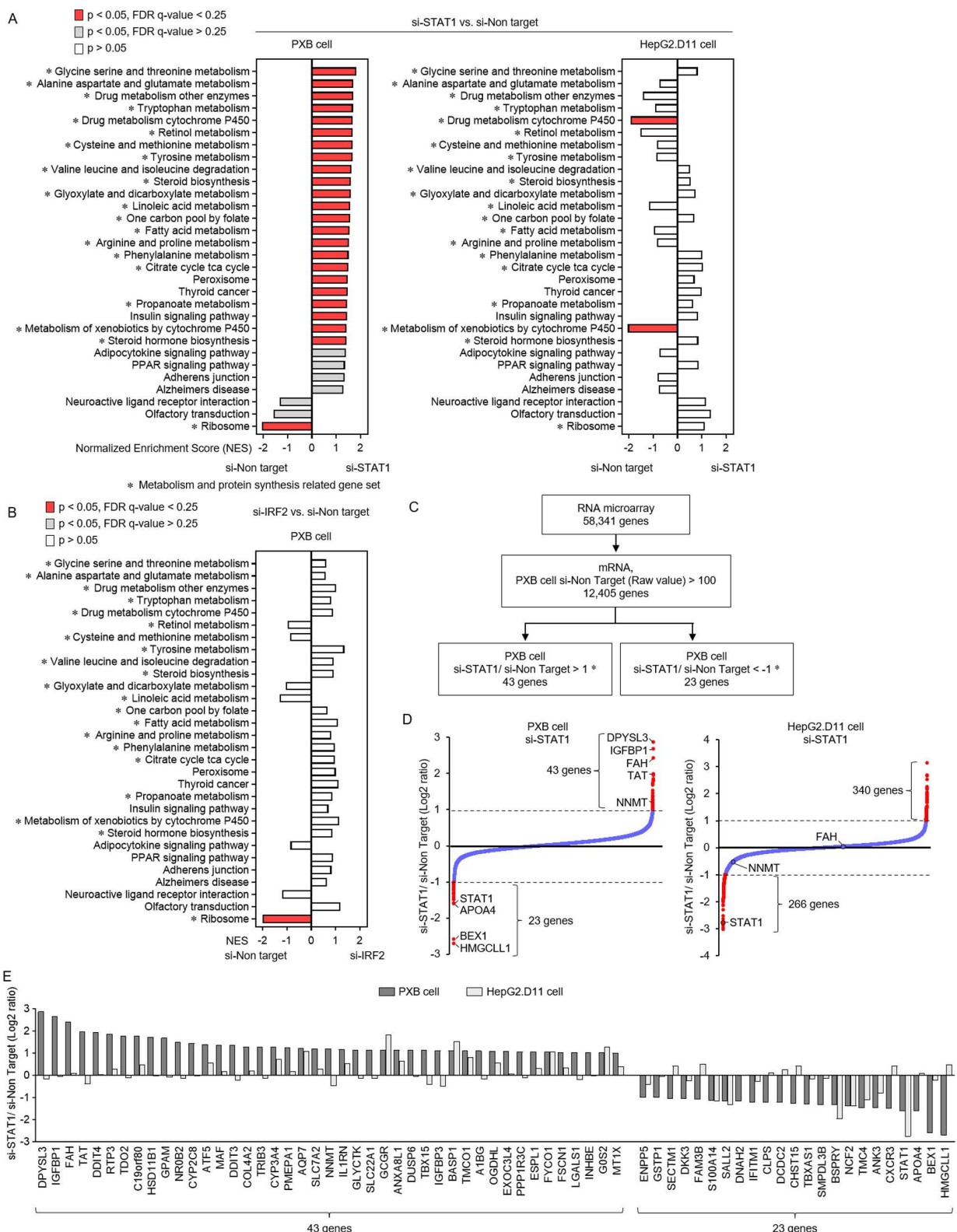

**Fig 2. RNA microarray analysis in knockdown of STAT1 in primary human hepatocyte HBV infection system and HBV-transfected HepG2 cells.** (A–E) RNA microarray was performed in HBV-infected PXB cells and HepG2.D11 cells with STAT1 knockdown and HBV-infected PXB cells with IRF2 knockdown. siRNA was transfected into the PXB cells on day 0 and HBV was added on day 4 and PXB cells were collected on day

13. HepG2.D11 cells were seeded on day 0 and siRNA was transfected at medium change on day 1. HepG2.D11 cells were collected on day 5. (A, B) Results of the Gene Set Enrichment Analysis (GSEA) of the Kyoto Encyclopedia of Genes and Genomes (KEGG) pathway. Bars in red indicate significant enrichment at FDR < 0.25, bars in gray represent gene sets with FDR > 0.25 and a nominal p value < 0.05 and bars in white represent gene sets with a nominal p value > 0.05. A positive normalized enrichment score (NES) value indicates enrichment in the si-STAT1 in PXB cells (left) and HepG2.D11 cells (right) (A) and si-IRF2 in PXB cells (B). * The gene sets related metabolism and protein synthesis. (C) The algorithm to extract genes changed by STAT1 knockdown in HBV-infected PXB cells. (D) Relative changes [si-STAT1/ si-Non-Target (Log2 ratio)] of 12,405 genes [PXB cell si-Non-Target (Raw value) > 100] in STAT1 knockdown in HBV-infected PXB cells and HepG2.D11 cells. (E) Relative changes of 63 genes (43 up- and 23 down-regulated genes) extracted by the algorithm in STAT1 knockdown of PXB cells and HepG2.D11 cells. Abbreviations: FDR, false discovery rate; GSEA, Gene Stet Enrichment Analysis; HBsAg, hepatitis B surface antigen; HBV, hepatitis B virus; IFN, interferon; IRF, interferon regulatory factor; KEGG, Kyoto Encyclopedia of Genes and Genomes; PXB, primary human hepatocyte; siRNA, small interfering RNA; STAT1, signal transducer and activator of transcription 1.

selected (including internal controls) [11,12,19–22]. DMF increased nuclear erythroid-related factor 2 (NRF2) in HepG2.D11 cells. Autophagy-related molecules, such as p62/sequestosome 1 (SQSTM1), autophagy-related gene (ATG)5, and ATG7, were upregulated by DMF. DMF induced antiviral molecules, including apolipoprotein B mRNA editing enzyme catalytic subunit (APOBEC)3A, APOBEC3B, and APOBEC3G, and interferon regulatory factor (IRF)3, IRF5, and IRF7. The anti-HBV molecules STAT1 and STAT2 were not induced by DMF. These results suggest that FAH promotes fumarate production and induces NRF2 and autophagy- and anti-HBV-related genes to exhibit antiviral effects in a STAT1/2-independent manner.

## Discussion

In the STAT1 knockdown experiment, the viral products were elevated in the conventional HBV-transfected HepG2 cells but decreased in the primary human hepatocyte infection system. RNA microarray analysis showed that STAT1 knockdown induced changes in intracellular molecules related to metabolism and protein synthesis in PXB cells, but not in HepG2. D11 cells. Comprehensive functional screening identified FAH as a candidate host factor that controls HBV infection independent of STAT1. DMF, the methyl ester of the FAH metabolite, showed antiviral effects and the expression of genes related to autophagy and anti-HBV effects were altered.

With the downregulation of STAT1, HBV gene expression was suppressed in primary human hepatocytes in an *in vitro* HBV infection assay system, whereas expression was enhanced in HepG2 cells transfected with the viral genome. Wilkening *et al.* performed a comparison of primary hepatocytes and hepatoma cell line HepG2 in the presence of different classes of promutagens [23]. The three promutagens caused DNA damage in primary human hepatocytes, but not in HepG2 cells. The most abundant isozyme of all P450s in the human liver, CYP3A4, is the most important isoform in drug metabolism in primary hepatocytes; however, CYP3A4 mRNA was not detected in HepG2 cells. In addition, the researchers detected mRNA P450 in primary hepatocytes, similar to that previously reported for human liver samples [24, 25]. Similarly, in this study, host gene expression in primary hepatocytes of the *in vitro* HBV infection assay system was markedly different from that in HepG2 cells with downregulation of STAT1.

A comprehensive functional screen of genes altered by STAT1 knockdown in PXB cells identified FAH as the gene exhibiting anti-HBV activity. FAH is the last enzyme in the tyrosine catabolic pathway [26] and catalyzes the breakdown of fumarylacetoacetate into fumarate and acetoacetate [27].

In this study, knockdown of FAH in HepG2.D11 cells resulted in a marked (25-fold) increase in intracellular HBV DNA levels. In contrast, the increase in intracellular HBV RNA levels was about 2-fold. In HBV life cycle, the pregenomic RNA (pgRNA) is packaged with

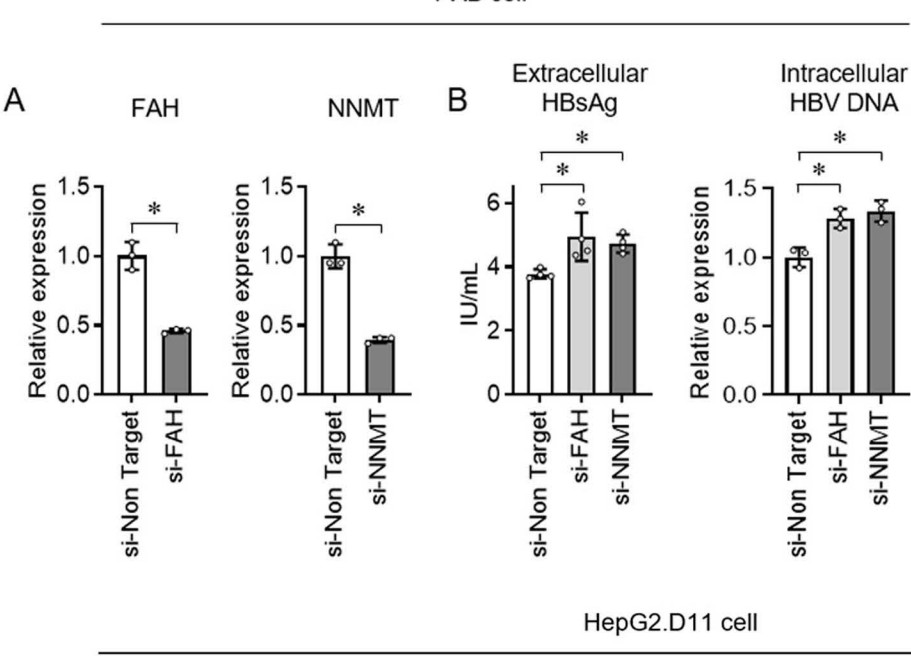

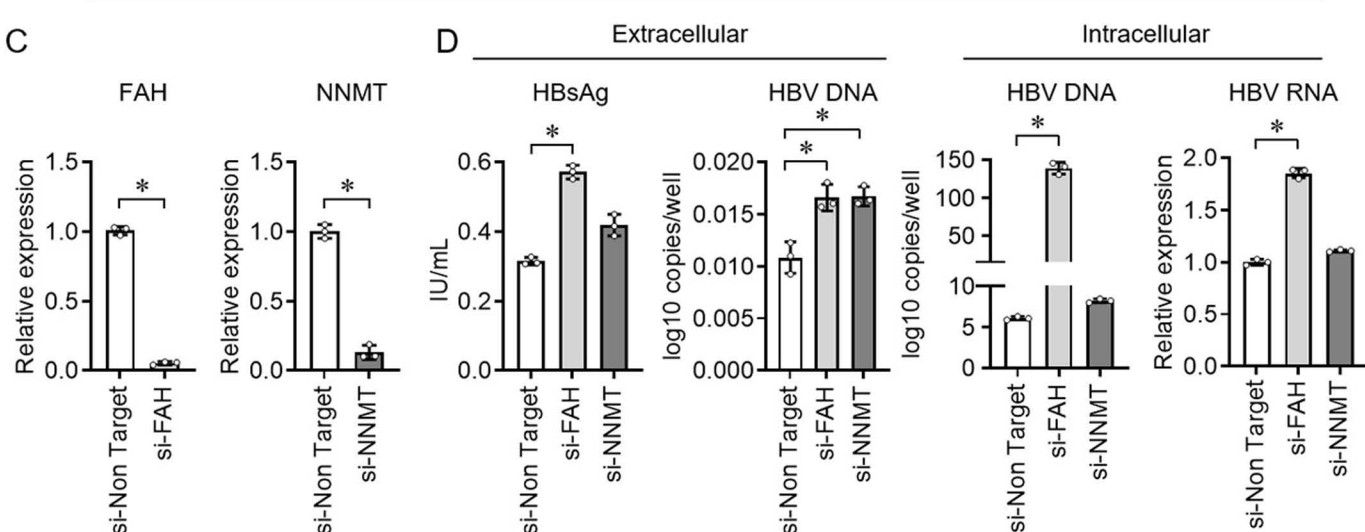

**Fig 3. Functional screen analysis of two candidate genes in HBV infection assay systems.** (A, B) siRNA of FAH and NNMT were transfected into the PXB cells on day 0 and HBV was added on day 4. Supernatant and PXB cells were collected on day 13. The culture medium was exchanged on day 4, 6, and 7. (A) mRNA expression of FAH and NNMT, and (B) extracellular hepatitis B surface antigen (HBsAg) and intracellular HBV DNA were analyzed on day 13. (C, D) HepG2. D11 cells were seeded on day 0 and siRNA of FAH and NNMT was transfected at medium change on day 1. HepG2.D11 cells were collected on day 5. (C) mRNA expression of FAH and NNMT and (D) extracellular HBsAg and HBV DNA and intracellular HBV DNA and HBV RNA were analyzed on day 5. (A–D) Data are represented as mean ± standard deviation (n = 3). (A, B) and (C, D) The experiments were replicated three times. (A, C) Mann-Whitney U test. (B, D) Tukey–Kramer post-hoc test. *$p < 0.05$. Abbreviations: FAH, fumarylacetoacetate hydrolase; HBsAg, hepatitis B surface antigen; HBV, hepatitis B virus; NNMT, nicotinamide N-methyltransferase; PXB, primary human hepatocyte; siRNA, small interfering RNA; STAT1, signal transducer and activator of transcription 1.

polymerase protein into immature nucleocapsids which consist of a core protein and are then reverse-transcribed into relaxed circular DNA (rcDNA) [28]. The variation in intracellular DNA and RNA levels upon FAH knockdown might indicate the possibility that FAH affects the pgRNA incorporation into the nucleocapsid and/or its reverse transcription process

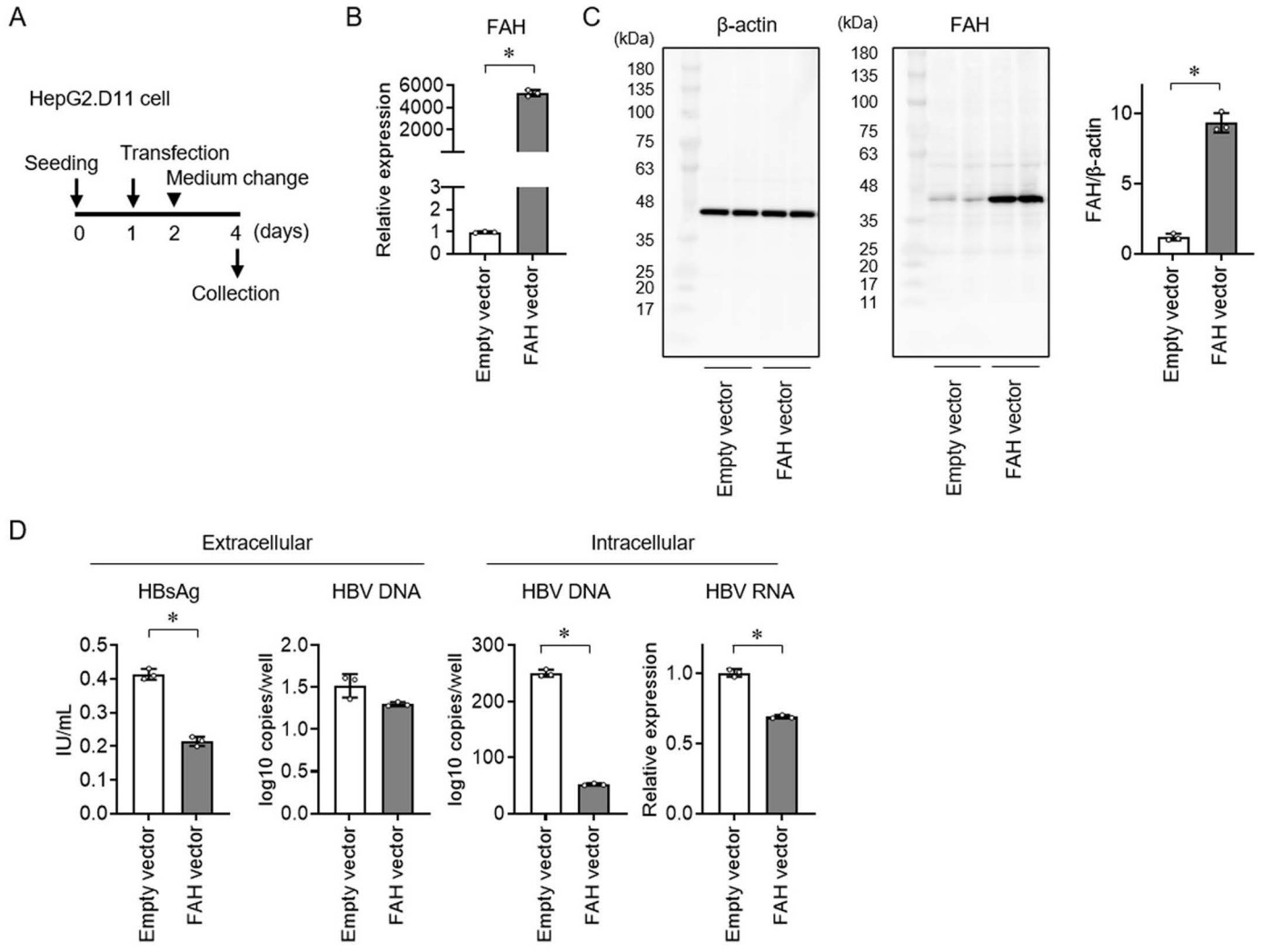

**Fig 4. Host gene FAH vector transfection in HBV-transfected HepG2 cells.** (A) HepG2.D11 cells were seeded on day 0 and CMV vector inserted cDNA FAH was transfected on day 1. The culture medium was exchanged on day 2 and the supernatant and HepG2.D11 cells were collected on day 4. (B) mRNA expression of FAH and (C) western blot analysis of extracts from HepG2.D11 using FAH antibody and normalized to β-actin on day 4. (D) Extracellular HBsAg and HBV DNA and intracellular HBV DNA and HBV RNA were analyzed on day 4. Data are represented as mean ± standard deviation (n = 3). The experiments were replicated three times. *p < 0.05 using the Mann-Whitney U test. Abbreviations: CMV, cytomegalovirus; FAH, fumarylacetoacetate hydrolase; HBsAg, hepatitis B surface antigen; HBV, hepatitis B virus.

during the HBV life cycle. Moreover, overexpression or knockdown of FAH in HepG2.D11 cells resulted in less pronounced changes in extracellular HBV DNA than in intracellular HBV DNA and HBV RNA or extracellular HBsAg. This is an important point for further functional evaluation of the FAH molecules extracted in this screening experiment, and we plan to clarify this point in a future experiment.

DMF, a fumarate derivative, is used as an immunomodulatory drug, especially in the treatment of multiple sclerosis [29]. In alcohol-related liver disease, DMF suppresses the inflammatory response and ameliorates hepatitis and lipidosis [30]. DMF may also be involved in cellular immune responses and antiviral defense mechanisms [31, 32].

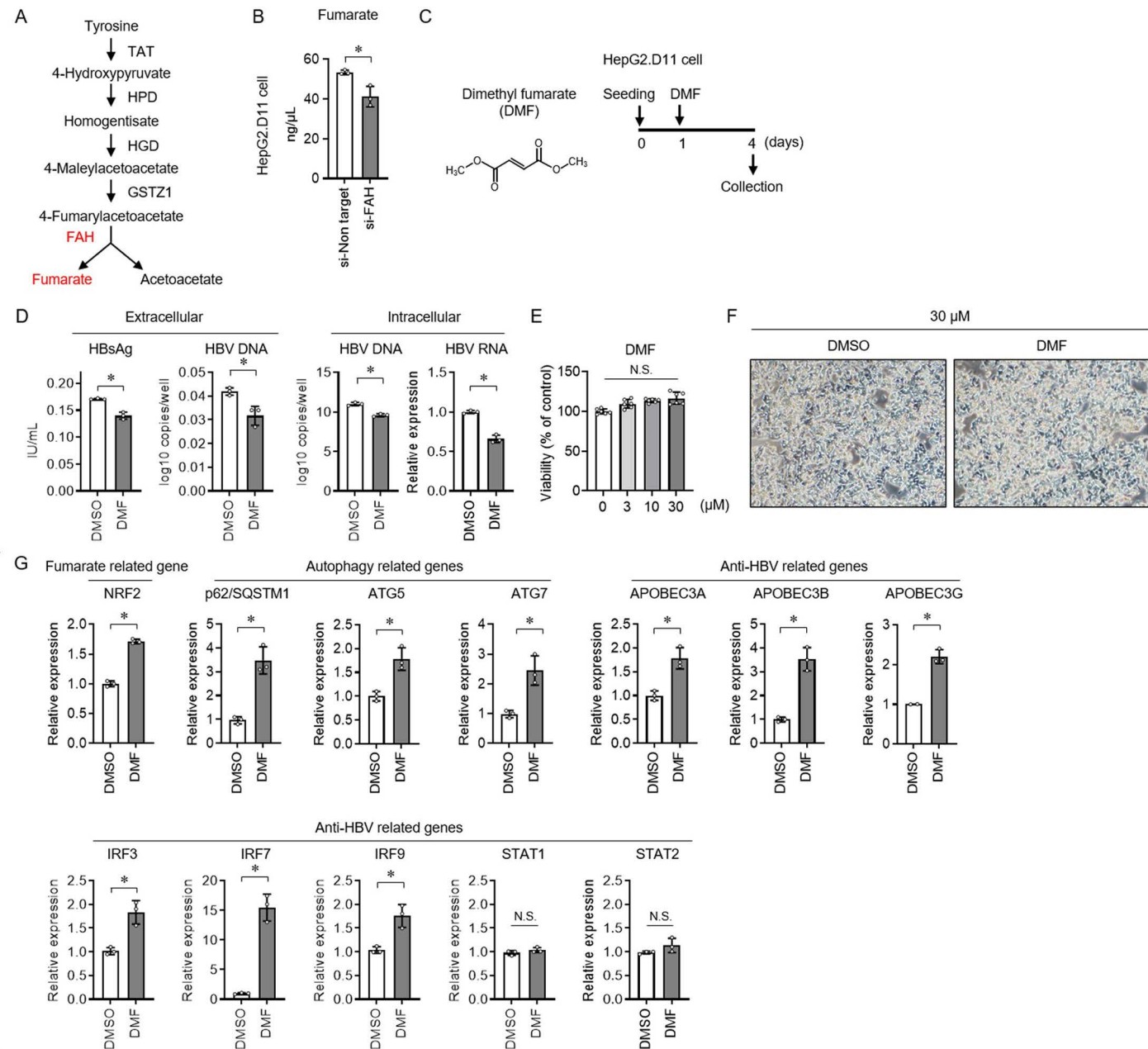

**Fig 5. Anti-HBV effect of dimethyl fumarate and gene expression analysis in hepatocytes.** (A) Diagram of the tyrosine metabolic pathway. The converting 4-fumarylacetoacetate to fumarate and acetoacetate is catalyzed by FAH. (B) HepG2.D11 cells were transfected with siRNA of FAH, and cells were collected 72 h later and intracellular fumaric acid concentration was measured. (C) Molecular structure of DMF (left). HepG2.D11 cells were seeded on day 0, DMF was added to the culture supernatant on day 1; supernatant and HepG2.D11 cells were collected on day 4 (right). (D) DMF and DMSO were added to HepG2.D11 cells at 30 μM and analyzed for extracellular HBsAg and HBV DNA and intracellular HBV DNA and HBV RNA on day 4. (E) DMF and DMSO were added to HepG2.D11 cells at 0, 3, 10, and 30 μM, and MTT assay was performed after 72 h to calculate cell viability. (F, G) DMF and DMSO were added to HepG2.D11 cells at 30 μM on day 1 and captured by phase contrast microscopy on day 4. Images were obtained from a ×4 objective (scale bar, 200 μm) (F). HepG2.D11 cells were collected on day 4 and mRNA expression was analyzed by real-time quantitative reverse transcription polymerase chain reaction (G). Data are represented as mean ± standard deviation (n = 3). (C-F) The experiments were replicated three times. *p < 0.05 using the Mann-Whitney U test.

DMF enhances the activity of NRF2, a transcription factor that regulates the expression of various antioxidant proteins and detoxification enzymes [33, 34]. NRF2 modulates p62/SQSTM1 which is a protein that targets ubiquitinated proteins for autophagic degradation, as well as autophagy initiating proteins such as ATG5/7 [19,35,36]. Autophagy plays an important role in HBV-related innate and adaptive immune responses [37]. Miyakawa *et al.* reported that galectin-9 suppresses HBV replication by selective autophagy of the viral core protein via p62/SQSTM1 [35,38]. In this study, DMF exhibited antiviral effects and increased NRF2 and autophagy-related gene expression. As shown in these papers, it is possible that autophagy may be involved in the antiviral effects of this study.

In this study, DMF also induced the expression of antiviral molecules such as APOBEC3A/B/G and IRF3/5/7 in a STAT1/2-independent manner. The APOBEC protein family, including APOBEC3A/B/G interferes with cccDNA stability through cytidine deamination and apurinic/apyrimidinic site formation [39–41]. The IRF family of transcription factors plays a critical role in the human innate immune response, with IFN production being a hallmark consequence of activation [42, 43]. These reports suggest that in this study, increased gene expression was associated with the anti-HBV effect of DMF. However, the exact gene signaling pathways and mechanisms in hepatocytes are unknown, and this is a great interest as the next subject.

In this study, we were focused on the influence of host factors on HBV transcription and amplification in hepatocytes. Transfection experiments in HepG2 cells were performed to examine intracellular viral transcription and amplification. We consider that there is no difference between HepG2 and HepG2-NTCP cells with respect to intracellular viral transcription and amplification. Using PXB cells, it is possible to observe viral infection. We performed experiments in which the order of HBV infection and siRNA knockdown was switched in the molecules we have examined, including STAT1. Gene knockdown did not affect the process of HBV entry into hepatocytes (S2 Fig). HBV infection experiments using HepG2-NTCP cells, which are not commercially available, are being considered as the next project.

In conclusion, knockdown of the host gene STAT1 showed antiviral activity in conventional HBV-transfected HepG2 cells, but conversely decreased viral replication in a primary human hepatocyte infection system. STAT1 knockdown induced changes in intracellular molecules related to metabolism and protein synthesis in PXB cells, but not in HepG2.D11 cells. FAH was identified as a candidate host factor that contributes to the control of viral infection independent of STAT1, and its metabolite, DMF, exhibited antiviral activity. These results suggest that the novel host factor, FAH, and its metabolites may be an innovative therapeutic strategy for controlling the HBV life cycle.

## Supporting information

**S1 Fig. Functional screen analysis of two candidate genes in HBV infection assay systems.** (A) siRNA of 43 candidate genes from the Cherry-pick library was transfected into the PXB cells on day 0 and HBV was added on day 4. The extracellular hepatitis B surface antigen (HBsAg) was analyzed on day 13. The culture medium was exchanged on days 4, 6, and 7. The 1.2-fold value of HBsAg in the sample transfected with the si-Non-Target is indicated by the dotted line. (B) Double siRNA of 12 target genes and STAT1 was transfected into the PXB cells on day 0 and HBV was added on day 4. The extracellular HBsAg and intracellular HBV DNA was analyzed on day 13. The 1.5-fold value of HBsAg in samples transfected with si-STAT1 is indicated by the dotted line. Abbreviations: HBsAg, hepatitis B surface antigen; HBV, hepatitis B virus; PXB, primary human hepatocyte; siRNA, small interfering RNA. (TIF)

**S2 Fig. An experiment in which the order of HBV infection and siRNA knockdown was reversed in PXB cells.** (A) HBV was added on day 0 and siRNA was transfected into the PXB cells on day 2. The supernatant and PXB cells were collected on day 13. The culture medium was exchanged on day 2, 4, and 7. (B) mRNA expression of STAT1 and (C) extracellular HBsAg and HBV DNA and intracellular HBV DNA and HBV RNA were analyzed on day 13. (TIF)

**S3 Fig.**
(TIF)

## Acknowledgments

The authors would like to thank Tomomi Kame, Masako Takada, and Kaori Yuki for their technical assistance, and Chieko Murata for assistance with manuscript preparation. We would like to thank Editage for the English language editing.

## Author contributions

**Conceptualization:** Takuto Nosaka, Yasunari Nakamoto.

**Data curation:** Takuto Nosaka, Tatsushi Naito, Yu Akazawa, Kazuto Takahashi, Hidetaka Matsuda, Masahiro Ohtani, Tsutomu Nishizawa, Hiroaki Okamoto, Yasunari Nakamoto.

**Funding acquisition:** Takuto Nosaka, Yasunari Nakamoto.

**Investigation:** Takuto Nosaka, Tatsushi Naito, Yu Akazawa, Kazuto Takahashi, Hidetaka Matsuda, Masahiro Ohtani, Tsutomu Nishizawa, Hiroaki Okamoto, Yasunari Nakamoto.

**Project administration:** Yasunari Nakamoto.

**Resources:** Takuto Nosaka, Tsutomu Nishizawa, Hiroaki Okamoto, Yasunari Nakamoto.

**Supervision:** Yasunari Nakamoto.

**Validation:** Takuto Nosaka, Yasunari Nakamoto.

**Visualization:** Takuto Nosaka, Yasunari Nakamoto.

**Writing – original draft:** Takuto Nosaka, Yasunari Nakamoto.

**Writing – review & editing:** Takuto Nosaka, Hiroaki Okamoto, Yasunari Nakamoto.

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
