## [Decision Letter · Decision Letter 0]

23 Aug 2024

PONE-D-24-31113Identification of novel antiviral host factors by functional gene expression analysis using in vitro HBV infection assay systemsPLOS ONE

Dear Dr. Nakamoto,

Thank you for submitting your manuscript to PLOS ONE. After careful consideration, we feel that it has merit but does not fully meet PLOS ONE’s publication criteria as it currently stands. Therefore, we invite you to submit a revised version of the manuscript that addresses the points raised during the review process. Please submit your revised manuscript by Oct 07 2024 11:59PM. If you will need more time than this to complete your revisions, please reply to this message or contact the journal office at plosone@plos.org . Please include the following items when submitting your revised manuscript:

We look forward to receiving your revised manuscript.

Kind regards,

Youkyung H. Choi, Ph.D.

Academic Editor

PLOS ONE

Journal Requirements:

When submitting your revision, we need you to address these additional requirements. 1. Please ensure that your manuscript meets PLOS ONE's style requirements, including those for file naming. The PLOS ONE style templates can be found at  https://journals.plos.org/plosone/s/file?id=wjVg/PLOSOne_formatting_sample_main_body.pdf and https://journals.plos.org/plosone/s/file?id=ba62/PLOSOne_formatting_sample_title_authors_affiliations.pdf 2. We noticed you have some minor occurrence of overlapping text with the following previous publication(s), which needs to be addressed: https://onlinelibrary.wiley.com/doi/10.1111/hepr.13722
https://onlinelibrary.wiley.com/doi/abs/10.1002/jmv.23916
https://www.dovepress.com/molecular-pharmacodynamics-of-new-oral-drugs-used-in-the-treatment-of--peer-reviewed-fulltext-article-DDDT
https://jbiomedsci.biomedcentral.com/articles/10.1186/s12929-023-00899-2
https://www.sciencedirect.com/science/article/pii/S0006295222001204?via%3Dihub In your revision ensure you cite all your sources (including your own works), and quote or rephrase any duplicated text outside the methods section. Further consideration is dependent on these concerns being addressed. 3. Thank you for stating the following financial disclosure:  "This research was partially supported by Japan Agency for Medical Research and Development (AMED) under Grant Numbers JP23fk0210104 and JP23fk0210113, and JSPS KAKENHI Grant-in-Aid for Scientific Research Number 22K15992." Please state what role the funders took in the study. If the funders had no role, please state: "The funders had no role in study design, data collection and analysis, decision to publish, or preparation of the manuscript."  If this statement is not correct you must amend it as needed.  Please include this amended Role of Funder statement in your cover letter; we will change the online submission form on your behalf. 4. Please expand the acronym “JSPS” (as indicated in your financial disclosure) so that it states the name of your funders in full. This information should be included in your cover letter; we will change the online submission form on your behalf. 5. Please note that funding information should not appear in the Acknowledgments section or other areas of your manuscript. We will only publish funding information present in the Funding Statement section of the online submission form. Please remove any funding-related text from the manuscript.  6. Please note that your Data Availability Statement is currently missing the direct link to access each database. If your manuscript is accepted for publication, you will be asked to provide these details on a very short timeline. We therefore suggest that you provide this information now, though we will not hold up the peer review process if you are unable. 7. PLOS ONE now requires that authors provide the original uncropped and unadjusted images underlying all blot or gel results reported in a submission’s figures or Supporting Information files. This policy and the journal’s other requirements for blot/gel reporting and figure preparation are described in detail at https://journals.plos.org/plosone/s/figures#loc-blot-and-gel-reporting-requirements and https://journals.plos.org/plosone/s/figures#loc-preparing-figures-from-image-files. When you submit your revised manuscript, please ensure that your figures adhere fully to these guidelines and provide the original underlying images for all blot or gel data reported in your submission.  See the following link for instructions on providing the original image data: https://journals.plos.org/plosone/s/figures#loc-original-images-for-blots-and-gels.    In your cover letter, please note whether your blot/gel image data are in Supporting Information or posted at a public data repository, provide the repository URL if relevant, and provide specific details as to which raw blot/gel images, if any, are not available. Email us at plosone@plos.org if you have any questions. 

**Additional Editor Comments:**

We invite you to submit a revised manuscript that addresses the points below. Specifically, different results from the infection and transfection experiments using PXB cells and HepG2-D11 cells need to be addressed. In addition, a rationale for HepG2 cells instead of HepG2-NTCP cells needs to be provided.

Reviewers' comments:

Reviewer's Responses to Questions

**Comments to the Author**

1. Is the manuscript technically sound, and do the data support the conclusions?

Reviewer #1: Yes

Reviewer #2: Partly

2. Has the statistical analysis been performed appropriately and rigorously? 

Reviewer #1: Yes

Reviewer #2: Yes

3. Have the authors made all data underlying the findings in their manuscript fully available?

Reviewer #1: Yes

Reviewer #2: Yes

4. Is the manuscript presented in an intelligible fashion and written in standard English?

Reviewer #1: No

Reviewer #2: Yes

5. Review Comments to the Author

Reviewer #1: The study by Nosaka et al. describes fumarylacetoacetate hydrolase (FAH) as a novel antiviral host factor against HBV. The authors found that siRNA-mediated knockdown of STAT1 unexpectedly reduced HBV replication in primary human hepatocytes (PXB cells). They then performed RNA microarray and identified 43 host genes that upregulated in the absence of STAT1 in PXB cells. Individual knockdown of these 43 genes revealed FAH as an antiviral host factor in PXB cells as well as in HepG2 cells. Furthermore, overexpression of FAH or addition of dimethyl fumarate (DMF), an FAH metabolite, reduced HBV replication in a STAT1-independent manner. The study is important since HBV remains uncurable and identifying new host factors may inform new antiviral strategies. However, the study contains several issues that need to be addressed.

1. The authors performed infection-based experiments in PXB cells and transfection-based experiments in HepG2 cells. This is not a fair comparison. The authors should use HepG2 cells stably expressing the HBV receptor NTCP and perform infection experiments in these cells so that the results from the two different cell culture systems can be compared.

2. Fig. 1, siRNA was transfected 4 days before infection, but the experiment lasted 13 days. Since siRNA-mediated gene knockdown is transient, the knockdown efficiency during infection day 4-13 is problematic which can be an issue in data interpretation.

3. Fig. 1D, the arrow for the day 1 label should be extended to the text above it. It’s confusing otherwise.

4. Figure 2B, I understand that the genes are chosen focused on PXB but indicating the number of genes that differ with treatment on the HepG2.D11 graph could be informative. Or point out the genes of interest for the other cell type on this graph too.

5. Figure 2C, asterisk is not explained.

6. Fig. 3, qPCR validation of the knockdowns is needed. Axes that say “relative expressions” should be “relative expression” singular. Lines labeled “folds” should say “fold” or “fold change” singular. Also, whether the red color represents significance is not explained.

7. Clarification that any comparisons not shown on graphs are non-significant would be useful.

8. In some graphs the authors indicate n=3 and show the mean. It would be better to show individual data points. If n= 3 means total data points (not 3 experimental replicates) the data pool is a little small.

9. Line 266, should “transfection” be “replication”?

10. Line 270, 1.2-fold is a rather marginal effect. Is this difference biologically meaningful?

11. Line 275, should “extracellular” be “intracellular”?

12. The sentence on lines 278-280 is confusing, I suggest it’s changed to "these results suggest that FAH has anti-HBV effects, as determined by siRNA screening of genes altered by STAT1 knockdown in PXB cells.”

13. Fig. 5F is quite zoomed out, I wonder if a slightly closer view would show cell morphology better.

14. Line 367, the sentence reads as if this is a conclusion, but the data do not really establish that DMF exerts its antiviral effect by inducing autophagy and anti-HBV related genes.

15. Line 371, what does “human biotransformation experiment” mean?

Reviewer #2: Authors of this manuscript investigated that fumarylacetoacetate hydrolase (FAH) is an antiviral host factor and methyl ester of FAH metabolite, dimethyl fumarate (DMF) has the antiviral effect in STAT1-independent pathway.

First of all, authors explained that knockdown of STAT1 increased viral products in HBV-transfected HepG2 cells therefore STAT1 is antiviral. And then decreased HBV replication and products in STAT1-KD, HBV-infected PXB cells is that HBV replication is reduced by STAT1-KD. The explanation by authors was not clear and the opposite results from the infection and transfection experiment should be thoroughly investigated.

To claim that STAT1 is antiviral, STAT1 overexpression should be conducted in HepG2-D11 cells with Western blotting with STAT1 and HBc and Northern and Southern blotting for HBV RNA and HBV DNA. If possible, also in PXB cells, too.

Since working with PXB cell may have some limitations, authors did not conduct Northern and Southern blotting for HBV RNA and DNA with PXB cells. Then, Northern and Southern blotting should be conducted in si-STAT-RNA transfected cells HepG2-D11 cell for clarification.

Again, Fig. 2 was a continuation of the opposite results between PXB and HepG2-D11 cells and the explanation was not clear.

In the Fig. 3A, FAH KD increased HBsAg in PXB cells and in the Fig 3B, FAH and STAT1 double KD increased HBsAg and intracellular DNA with decreased HBV DNA and HBsAg in STAT1 single KD. In should be included with FAH single KD in Fig 3B (right graph) to compare intracellular DNA level.

In Fig. 3D, intracellular DNA level was 25 times higher in FAH KD HepG2-D11 cells than control KD cells then why RNA level was only 2 times higher and extracellular DNA level was only increased little compared to intracellular DNA? Southern blotting for intracellular DNA and extracellular DNA should be conducted. Northern blotting for RNA should be conducted to compare with pgRNA, S and X mRNAs. At the same time, RT-qPCR for pgRNA only and for total RNA should be conducted.

Again, in Fig 4D in FAH overexpression, compare to HBV RNA, intracellular DNA level was decreased so much and extracellular DNA level was not decreased significantly. Southern blotting to compare RC DNA and DL DNA levels and Northern blotting for pgRNA, S and X mRNAs are needed.

Since DMF increased NRF2 expression (Fig 5G) and NRF2 regulate HMOX-1, then why HMOX-1 is not increased? It needs to be explained and/or speculated why so.

Minor points,

In lane 64, cccDNA drug resistance should be corrected to cccDNA persistence or else.

In lanes 141, ‘the selected target genes’ and 96 primers were used. How these gene were selected? I cannot find in the manuscript. And primer information was hard to find (Table 2).

In lanes 189, radioimmunoprecipitation assay (RIPA) lysis buffer was used for Western blotting. It is better to show the composition of the buffer, since there are many versions.

Lanes 407-409, it was not clear.

Lanes 421-423, this conclusion is not clear, which I already mentioned above.

6. PLOS authors have the option to publish the peer review history of their article (what does this mean? ). If published, this will include your full peer review and any attached files.

**Do you want your identity to be public for this peer review?** For information about this choice, including consent withdrawal, please see our Privacy Policy .

Reviewer #1: No

Reviewer #2: No

---

## [Author Response · Author response to Decision Letter 1]

9 Oct 2024

Reviewer #1:

1. The authors performed infection-based experiments in PXB cells and transfection-based experiments in HepG2 cells. This is not a fair comparison. The authors should use HepG2 cells stably expressing the HBV receptor NTCP and perform infection experiments in these cells so that the results from the two different cell culture systems can be compared.

We appreciate your important suggestion. As you pointed out, it is not possible to observe HBV entry in HepG2 cells.

In this study, we were focused on the influence of host factors on HBV transcription and amplification in hepatocytes. Transfection experiments in HepG2 cells were performed to examine intracellular viral transcription and amplification. We consider that there is no difference between HepG2 and HepG2-NTCP cells with respect to intracellular viral transcription and amplification. Using PXB cells, it is possible to observe viral infection. We performed experiments in which the order of HBV infection and siRNA knockdown was switched in the molecules we have examined, including STAT1. Gene knockdown did not affect the process of HBV entry into hepatocytes (S2 Fig). HBV infection experiments using HepG2-NTCP cells, which are not commercially available, are being considered as the next project.

Accordingly, we added the description to the Discussion section (page 31-32, lines 459-468) and Figure and legend of S2 Fig.

2. Fig. 1, siRNA was transfected 4 days before infection, but the experiment lasted 13 days. Since siRNA-mediated gene knockdown is transient, the knockdown efficiency during infection day 4-13 is problematic which can be an issue in data interpretation.

We appreciate your comments. Compared to HepG2 cells, which are a conventional cell line, PXB cells are the primary hepatocytes which do not proliferate, and we are considering the possibility of long-term siRNA knockdown effect. Though the course of the knockdown effect is not known, however, we have confirmed that the knockdown effect is still sufficient at day 13 (Fig 1B).

3. Fig. 1D, the arrow for the day 1 label should be extended to the text above it. It’s confusing otherwise.

We appreciate your suggestion. The arrow for the day 1 label was extended to the text above it (Fig 1D).

4. Figure 2B, I understand that the genes are chosen focused on PXB but indicating the number of genes that differ with treatment on the HepG2.D11 graph could be informative. Or point out the genes of interest for the other cell type on this graph too.

We appreciate your important suggestion. We indicated the number of genes that were altered in si-STAT1 in HepG2.D11 cells.

In HepG2.D11 cells, STAT knockdown increased expression of 340 genes [si-STAT1/si-Non-Target (Log2 ratio) > 1] and decreased expression of 266 genes [si-STAT1/si-Non-Target (Log2 ratio) < -1] (Fig 2D).

Accordingly, we added the description to the Results section (page 22, lines 276-278) and modified Fig 2D.

5. Figure 2C, asterisk is not explained.

We appreciate your comments. The asterisk in Figure 2E (before 2C) was a typographical error and removed.

6. Fig. 3, qPCR validation of the knockdowns is needed. Axes that say “relative expressions” should be “relative expression” singular. Lines labeled “folds” should say “fold” or “fold change” singular. Also, whether the red color represents significance is not explained.

We appreciate your important suggestion. Knockdown experiments were performed in PXB cells using siRNA. The results for two genes, FAH and NNMT, in which knockdown efficiency was confirmed by qRT-PCR, are shown in Fig. 3A and B. The expression of the target molecule was significantly reduced by knockdown experiment. Results for other siRNAs were provided in Supplemental Figure 1.

Axes that say “relative expressions” were modified to “relative expression” singular in Figure 3, 4, and 5. Lines labeled “folds” were modified to “fold change” singular in Supplemental Figure 1. The description that red color does not represent significance is described in legend of Supplemental Figure 1.

7. Clarification that any comparisons not shown on graphs are non-significant would be useful.

We appreciate your comments. We clarified that any comparisons not shown on graphs are non-significant in Materials and Methods section (page 19, lines 211-212).

8. In some graphs the authors indicate n=3 and show the mean. It would be better to show individual data points. If n= 3 means total data points (not 3 experimental replicates) the data pool is a little small.

We appreciate your important suggestion. The number of times the experiment was replicated is described in the figure legends.

9. Line 266, should “transfection” be “replication”?

We appreciate your comments. We changed “transfection” to “replication” (page 24, line 306).

10. Line 270, 1.2-fold is a rather marginal effect. Is this difference biologically meaningful?

We appreciate your comments. A 1.2-fold increase is a rather marginal effect, and this difference is not considered to have biologically meaningful. Unfortunately, it was not so easy to show large increases in the amount of HBV, especially in HBV-transfected cells. Considering the characteristics of HBV stable transfection cells, we set the fold change at 1.2 in order to select the genes that have an anti-HBV effect from among the 43 candidate genes.

11. Line 275, should “extracellular” be “intracellular”?

We appreciate your comments. We changed “extracellular” to “intracellular” (page 24, line 315).

12. The sentence on lines 278-280 is confusing, I suggest it’s changed to "these results suggest that FAH has anti-HBV effects, as determined by siRNA screening of genes altered by STAT1 knockdown in PXB cells.”

We appreciate your suggestion. The sentence was changed to "These results suggest that FAH has anti-HBV effects, as determined by siRNA screening of genes altered by STAT1 knockdown in PXB cells.” (page 25, lines 321-322)

13. Fig. 5F is quite zoomed out, I wonder if a slightly closer view would show cell morphology better.

We appreciate your suggestion. The image of Fig 5F was enlarged to improve the visibility of the cell morphology.

14. Line 367, the sentence reads as if this is a conclusion, but the data do not really establish that DMF exerts its antiviral effect by inducing autophagy and anti-HBV related genes.

We appreciate your suggestion. As you pointed out, the data do not really establish that DMF exerts its antiviral effect by inducing autophagy and anti-HBV related genes. The sentence was changed to “DMF, the methyl ester of the FAH metabolite, showed antiviral effects and the expression of genes related to autophagy and anti-HBV effects were altered.” (page 29, lines 406-408)

15. Line 371, what does “human biotransformation experiment” mean?

We appreciate your suggestion. The words “human biotransformation experiment” were removed. (page 29, line 411)

Reviewer #2:

First of all, authors explained that knockdown of STAT1 increased viral products in HBV-transfected HepG2 cells therefore STAT1 is antiviral. And then decreased HBV replication and products in STAT1-KD, HBV-infected PXB cells is that HBV replication is reduced by STAT1-KD. The explanation by authors was not clear and the opposite results from the infection and transfection experiment should be thoroughly investigated.

We appreciate your important suggestion. As you pointed out, in HBV-transfected HepG2 cells, knockdown of STAT1 increased viral products. And in HBV-infected PXB cells, knockdown of STAT1 decreased HBV replication.

To investigate the mechanism by which STAT1 knockdown reduces HBV levels in primary human hepatocytes, RNA microarray analysis was performed and to comprehensively examine the molecular changes in PXB cells. In HBV-infected PXB cells, STAT1 knockdown enriched 24 gene sets (p < 0.05, FDR q-value < 0.25) (Fig 2A). Of the 24 gene sets, 21 were associated with metabolism and protein synthesis. However, STAT1 knockdown in HepG2.D11 cells and IRF2 knockdown in PXB cells did not result in similar molecular changes (Fig 2B). Accordingly, we added the description to the Results section (page 22, lines 266-272), Figures and legends of Fig 2A and B, and the Materials and Methods section (page 15, lines 162-168).

Moreover, the following text is described in the Discussion section as a mechanism for the changes in the signal pathways related to metabolism and protein synthesis in the PXB cell infection system, but not in the HepG2 cell transfection system.

Wilkening et al. performed a comparison of primary hepatocytes and hepatoma cell line HepG2 in the presence of different classes of promutagens [22]. The three promutagens caused DNA damage in primary human hepatocytes, but not in HepG2 cells. The most abundant isozyme of all P450s in the human liver, CYP3A4, is the most important isoform in drug metabolism in primary hepatocytes; however, CYP3A4 mRNA was not detected in HepG2 cells. In addition, the researchers detected mRNA P450 in primary hepatocytes, similar to that previously reported for human liver samples [23, 24]. Similarly, in this study, host gene expression in primary hepatocytes of the in vitro HBV infection assay system was markedly different from that in HepG2 cells with downregulation of STAT1. A comprehensive functional screen of genes altered by STAT1 knockdown in PXB cells identified FAH as the gene exhibiting anti-HBV activity. FAH is the last enzyme in the tyrosine catabolic pathway [25] and catalyzes the breakdown of fumarylacetoacetate into fumarate and acetoacetate [26]. (page 29-30, lines 411-424).

To claim that STAT1 is antiviral, STAT1 overexpression should be conducted in HepG2-D11 cells with Western blotting with STAT1 and HBc and Northern and Southern blotting for HBV RNA and HBV DNA. If possible, also in PXB cells, too. Since working with PXB cell may have some limitations, authors did not conduct Northern and Southern blotting for HBV RNA and DNA with PXB cells. Then, Northern and Southern blotting should be conducted in si-STAT-RNA transfected cells HepG2-D11 cell for clarification.

We appreciate your suggestion. STAT1 overexpression was conducted in HepG2.D11 cells.

To overexpress STAT1 in HepG2.D11 cells, STAT1 vector was transfected (Fig 1G). Increased STAT1 expression was confirmed by quantitative real-time polymerase chain reaction (PCR) (Fig 1H) and western blot (Fig 1I). Overexpression of STAT1 decreased extracellular HBsAg and HBV DNA and intracellular HBV DNA and RNA levels (Fig 1J). Accordingly, we added the description to the Results section (page 20, lines 231-235), Figures and legends of Fig 1G-J, and the Materials and Methods section (page 18, lines 181-186 and page 19, lines 199-206).

In addition, we conducted transfection experiments using STAT1 and FAH vectors in primary human hepatocytes, PXB cells, however, functional overexpression of the target molecule was not possible. Unlike conventional cell lines, PXB cells do not proliferate, so the amount of RNA and DNA recovered is small. Thus, we did not conduct Northern blotting or Southern blotting.

As you suggested, Northern and Southern blotting should be conducted in si-STAT-RNA transfected cells HepG2.D11 cell for clarification. We plan to perform Northern and Southern blotting for HBV RNA and DNA in si-STAT-RNA transfected HepG2-D11 cells in three months, and the results will be published in an upcoming paper summarized below.

< Summary of the upcoming paper; UNPUBLISHED and CONFIDENTIAL >

Title: Functional Analysis of Fumarylacetoacetate Hydrolase (FAH) as a Hepatocyte Host Factor Gene Regulating Hepatitis B Virus Replication and Hepatocarcinogenesis

Background: Recent molecular biological analyses have highlighted the importance of host factors in anti-viral and/or anti-tumor effects. We established a hepatitis B virus (HBV) infection assay system using primary human hepatocytes and extracted fumarylacetoacetate hydrolase (FAH) as a host factor gene that regulates viral life cycle by RNA microarray analysis (in submission). In this study, we investigated the anti-HBV and anti-tumor effects of FAH using resected hepatocellular carcinoma (HCC) tissue and HBV stable transfected HCC cell lines.

Methods: In 60 resected HCC specimens, the intensity of immunohistological staining of FAH in tumor and non-tumor areas was determined by H-score using inform (akoya), and the correlation with prognosis was analyzed. The FAH index was calculated by H-score of tumor/non-tumor area. The 1.3 mer HBV genome of genotype C2 with basic core promoter and/or pre-core mutation (wt/wt; Low transcription, mu/mu; High t.) was transfected into HepG2 cells and HepG2.1-E10 (Low t.) and HepG2.D11 (High t.) were established. FAH vector and siRNA were transfected and MTT assay, apoptosis assay, and sphere formation assay were performed. The HBsAg was standardized by cell count (HBsAg index). RNA expression was analyzed for 205 anti-tumor and anti-viral related genes using TaqMan custom array plates.

Results: The H-score of FAH in resected HCC specimens was significantly lower in tumor areas compared to non-tumor areas (tumor/non-tumor areas: 143/165). Recurrence and survival rates were worse in the　patient group with lower FAH index (p < 0.05). In HepG2.D11 and HepG2.1-E10, the viability index and HBsAg index were increased in the FAH knockdown and decreased in the FAH vector transduction experiments compared to controls. Furthermore, FAH overexpression in HepG2.D11 and HepG2.1-E10 induced apoptosis and decreased the number of spheres. In FAH overexpression, the expression of CCKN2B and CHEK1 (cell cycle) were elevated, NANOG and SALL4 (stem cell function) were decreased, and anti-HBV related genes, APOBEC3B and IRF7, were increased.

Conclusions: Evaluation of FAH expression in HCC tissues and the functional analysis in HBV-transfected HepG2 cells suggest that the FAH gene may contribute to HBV reduction, inhibition of cell proliferation, and induction of cell apoptosis. Hepatocyte host factor FAH is shown to have the potential to exhibit both anti-tumor and anti-HBV effects, and may be a candidate for a new therapeutic strategy of viral carcinogenesis.

Again, Fig. 2 was a continuation of the opposite results between PXB and HepG2-D11 cells and the explanation was not clear.

We appreciate your suggestion. The opposite results between PXB and HepG2-D11 cells were observed. Thus, to investigate the mechanism by which STAT1 knockdown reduces HBV levels in primary human hepatocytes, RNA microarray analysis was performed and to comprehensively examine the molecular changes in PXB cells. In HBV-infected PXB cells, STAT1 knockdown enriched 24 gene sets (p < 0.05, FDR q-value < 0.25) (Fig 2A). Of the 24 gene sets, 21 were associated with metabolism and protein synthesis. However, STAT1 knockdown in HepG2.D11 cells and IRF2 knockdown in PXB cells did not result in similar molecular changes (Fig 2B). Accordingly, we added the description to the Results section (page 22, lines 266-272), Figures and legends of Fig 2A and B, and the Materials and Methods section (page 15, lines 162-168).

In the Fig. 3A, FAH KD increased HBsAg in PXB cells and in the Fig 3B, FAH and STAT1 double KD increased HBsAg and intracellular DNA with decreased HBV DNA and HBsAg in STAT1 single KD. In should be included with FAH single KD in Fig 3B (right graph) to compare intracellular DNA level.

We appreciate your suggestion. Intracellular HBV DNA level in FAH single KD was shown in Fig 3B.

Knockdown of FAH and NNMT significantly elevated extracellular HBsAg and intracellular HBV DNA levels (Fig 3B).

Accordingly, we added the description to the Results section (page 24, lines 316-318), Figure and legend of Fig 3B.

In Fig. 3D, intracellular DNA level was 25 times higher in FAH KD HepG2-D11 cells than control KD cells then why RNA level was only 2 times higher and extracellular DNA level was only increased little compared to intracellular DNA? Southern blotting for intracellular DNA and extracellular DNA should be conducted. Northern blotting for RNA should be conducted to compare with pgRNA, S and X mRNAs. At the same time, RT-qPCR for pgRNA only and for total RNA should be conducted.

Again, in Fig 4D in FAH over

---

## [Decision Letter · Decision Letter 1]

7 Nov 2024

PONE-D-24-31113R1Identification of novel antiviral host factors by functional gene expression analysis using in vitro HBV infection assay systemsPLOS ONE

Dear Dr. Nakamoto,

Thank you for submitting your manuscript to PLOS ONE. After careful consideration, we feel that it has merit but does not fully meet PLOS ONE’s publication criteria as it currently stands. Therefore, we invite you to submit a revised version of the manuscript that addresses the points raised during the review process.

The revised manuscript improved to make better understanding.

However, a few things need to be corrected.

Figures 1, 3, and S1 still contain plural descriptions in graph description and line labeling.

A reference in line 164 needs to be formatted as other ones. 

We look forward to receiving your revised manuscript.

Kind regards,

Youkyung H. Choi, Ph.D.

Academic Editor

PLOS ONE

Journal Requirements:

Reviewers' comments:

Reviewer's Responses to Questions

**Comments to the Author**

1. If the authors have adequately addressed your comments raised in a previous round of review and you feel that this manuscript is now acceptable for publication, you may indicate that here to bypass the “Comments to the Author” section, enter your conflict of interest statement in the “Confidential to Editor” section, and submit your "Accept" recommendation.

Reviewer #1: All comments have been addressed

2. Is the manuscript technically sound, and do the data support the conclusions?

Reviewer #1: Yes

3. Has the statistical analysis been performed appropriately and rigorously? 

Reviewer #1: Yes

4. Have the authors made all data underlying the findings in their manuscript fully available?

Reviewer #1: Yes

5. Is the manuscript presented in an intelligible fashion and written in standard English?

Reviewer #1: Yes

6. Review Comments to the Author

Reviewer #1: (No Response)

7. PLOS authors have the option to publish the peer review history of their article (what does this mean? ). If published, this will include your full peer review and any attached files.

**Do you want your identity to be public for this peer review?** For information about this choice, including consent withdrawal, please see our Privacy Policy .

Reviewer #1: No

---

## [Author Response · Author response to Decision Letter 2]

9 Nov 2024

Reviewer #1:

The revised manuscript improved to make better understanding. However, a few things need to be corrected. Figures 1, 3, and S1 still contain plural descriptions in graph description and line labeling. A reference in line 164 needs to be formatted as other ones.

We appreciate your important suggestion. We revised plural descriptions in graph description and line labeling in Figures 1, 3, and S1. We formatted a reference in line 164 as other ones.

---

## [Editor Report · Decision Letter 2]

13 Nov 2024

Identification of novel antiviral host factors by functional gene expression analysis using in vitro HBV infection assay systems

PONE-D-24-31113R2

Dear Dr. Nakamoto,

We’re pleased to inform you that your manuscript has been judged scientifically suitable for publication and will be formally accepted for publication once it meets all outstanding technical requirements.

Kind regards,

Youkyung H. Choi, Ph.D.

Academic Editor

PLOS ONE
---

## [Editor Report · Acceptance letter]

PONE-D-24-31113R2

PLOS ONE

Dear Dr. Nakamoto,

I'm pleased to inform you that your manuscript has been deemed suitable for publication in PLOS ONE. Congratulations! Your manuscript is now being handed over to our production team.

Kind regards,

on behalf of

Dr. Youkyung H. Choi

Academic Editor

PLOS ONE